# Feasibility and Acceptability of a Complex Telerehabilitation Intervention for Pediatric Acquired Brain Injury: The Child in Context Intervention (CICI)

**DOI:** 10.3390/jcm11092564

**Published:** 2022-05-03

**Authors:** Ingvil Laberg Holthe, Nina Rohrer-Baumgartner, Edel J. Svendsen, Solveig Lægreid Hauger, Marit Vindal Forslund, Ida M. H. Borgen, Hege Prag Øra, Ingerid Kleffelgård, Anine Pernille Strand-Saugnes, Jens Egeland, Cecilie Røe, Shari L. Wade, Marianne Løvstad

**Affiliations:** 1Department of Research, Sunnaas Rehabilitation Hospital, 1453 Nesodden, Norway; uxronb@sunnaas.no (N.R.-B.); edel.jannecke.svendsen@sunnaas.no (E.J.S.); solveig.laegreidhauger@sunnaas.no (S.L.H.); hege.ora@sunnaas.no (H.P.Ø.); marianne.lovstad@sunnaas.no (M.L.); 2Department of Psychology, Faculty of Social Sciences, University of Oslo, 0317 Oslo, Norway; idmbor@ous-hf.no (I.M.H.B.); jens.egeland@siv.no (J.E.); 3Department of Nursing and Health Promotion, Oslo Metropolitan University, 0130 Oslo, Norway; 4Research Centre for Habilitation and Rehabilitation Models and Services (CHARM), Institute of Health and Society, Faculty of Medicine, University of Oslo, 0372 Oslo, Norway; cecilie.roe@medisin.uio.no; 5Department of Physical Medicine and Rehabilitation, Oslo University Hospital, 0424 Oslo, Norway; mavfor@ous-hf.no (M.V.F.); uxinff@ous-hf.no (I.K.); 6Department of Acquired Brain Injury, Statped: Norwegian Service for Special Needs Education, 2815 Gjøvik, Norway; anine.strand-saugnes@statped.no; 7Department of Research, Vestfold Hospital Trust, 3103 Tønsberg, Norway; 8Institute of Clinical Medicine, Faculty of Medicine, University of Oslo, 0372 Oslo, Norway; 9Division of Pediatric Rehabilitation Medicine, Cincinnati Children’s Hospital Medical Center, Cincinnati, OH 45229, USA; shari.wade@cchmc.org; 10Department of Pediatrics, University of Cincinnati College of Medicine, Cincinnati, OH 45267, USA

**Keywords:** feasibility study, goal-oriented rehabilitation, pediatric brain injury, SMART-goals, home-based rehabilitation

## Abstract

The current study is a feasibility study of a randomized controlled trial (RCT): the Child in Context Intervention (CICI). The CICI study is an individualized, goal-oriented and home-based intervention conducted mainly through videoconference. It targets children with ongoing challenges (physical, cognitive, behavioral, social and/or psychological) after acquired brain injury (ABI) and their families at least one year post injury. The CICI feasibility study included six children aged 11–16 years with verified ABI-diagnosis, their families and their schools. The aim was to evaluate the feasibility of the intervention components, child and parent perceptions of usefulness and relevance of the intervention as well as the assessment protocol through a priori defined criteria. Overall, the families and therapists rated the intervention as feasible and acceptable, including the videoconference treatment delivery. However, the burden of assessment was too high. The SMART-goal approach was rated as useful, and goal attainment was high. The parents’ ratings of acceptability of the intervention were somewhat higher than the children’s. In conclusion, the CICI protocol proved feasible and acceptable to families, schools and therapists. The assessment burden was reduced, and adjustments in primary outcomes were made for the definitive RCT.

## 1. Introduction

Although children with acquired brain injury (pediatric acquired brain injury—pABI) may suffer long-lasting physical, cognitive, behavioral and social symptoms as well as disturbances in psychological adaptation [1,2,3], few randomized controlled trials have explored the effectiveness of complex interventions targeting these children, their families and their everyday context at home and in school. This may be due to the considerable complexity of designing and conducting such studies where one must take a range of factors into account. These include the child’s age and developmental level, the heterogeneity of symptoms that may follow a pABI, family factors that may influence the ability and willingness to participate in an intervention, and the heterogeneity in factors related to the child’s everyday life in the community, such as school and other areas of participation. The possible pitfalls are many. Exploring the feasibility of a complex intervention is therefore important prior to large-scale RCTs. This paper describes the investigation of the feasibility of a complex, individualized and community-based intervention targeting children with pABI and their families.

Brain damage acquired after birth can be caused by traumatic (TBI) or non-traumatic injuries such as stroke (i.e., brain hemorrhage or infarction), tumor, inflammation, infection or hypoxia. pABI may cause a wide range of disruptions in the child’s developmental course [1,2,3]. Deficits may be persistent, leading to reduced participation in school and social life [4,5,6], and reduced quality of life [2,7,8]. Prior studies have identified needs for long-term follow-up after ABI; psychological support for parents, siblings and the affected child; as well as support in the return to school [9]. Interestingly, reported needs may change over time, from primarily physical concerns in the first months post injury (79%), to cognitive (47%) and socioemotional needs (68%) later on [10]. Importantly, rehabilitation needs after pABI are described as largely unmet [11].

As family needs after a pediatric brain injury are complex and heterogeneous, rehabilitation services and follow-up care should be individualized to each patient’s needs. In line with this, rehabilitation should be multidisciplinary and take place within a bio-psychosocial framework [12]. In addition, rehabilitation needs to be flexible, monitoring the development and goals of the patient, which may change over time [12]. A recent study mapped treatment goals set by young people with ABI to the domains in the International Classification of Functioning, Disability and Health (ICF) [13], in which 52% were related to activities and participation, and 20% to environmental factors [14]. This highlights the need to include community-based rehabilitation that is individualized and context-sensitive [14]. The complexity of pediatric ABI rehabilitation may explain the small body of evidence-based recommendations. However, existing knowledge provides evidence for family/caregiver-focused interventions and the use of technology in rehabilitation [15]. In addition, direct interventions related to cognitive functions such as attention, memory, executive function and emotional/behavioral functioning have given positive results [15]. However, a critique against the direct training approach in cognitive rehabilitation (i.e., interventions directed toward retraining the child’s abilities) is that generalization to everyday functioning is often unknown or disputed [16]. Therefore, indirect approaches such as behavioral compensation and modification, environmental modification and supports, educational supports and instructional strategies are recommended, including education for parents and caregiver involvement in handling executive dysfunction [17,18]. The effects of parenting/family interventions after pABI have been explored in randomized controlled trials (RCTs) by Wade and colleagues with promising results for online problem-solving training, improving behavior problems, executive functioning, and family functioning [15,19,20,21,22]. The use of technology in rehabilitation is relatively new, but telerehabilitation has shown to be a reliable alternative to in-person meetings [23,24] with promising results in cognitive rehabilitation [25], speech and language therapy [26] and family therapy [27,28]. Recent reviews on technology-based or -assisted rehabilitation programs for children with ABI have found that in addition to the mentioned evidence for online problem-solving, there is promising evidence for training of cognitive, social and behavioral skills [15,22,29,30]. The findings are characterized by heterogeneity and small samples, but the involvement of a clinician in addition to the technology-based intervention may be of importance for a positive result [30]. Although most of this research has been conducted in the USA, some of the technology-based intervention studies presented in the literature reviews have also been performed in Europe. In addition, goal-oriented rehabilitation is considered a key ingredient in pABI rehabilitation [14,31], and goal attainment scaling can be a sensitive and meaningful way to measure rehabilitation outcomes [32]. Children’s self-identified goals have been found to be as achievable as their parents’ goals in a sample of children with a mean age of 9 years [33], pointing toward the importance of including the children in defining rehabilitation goals.

To our knowledge, no single intervention has included all these important perspectives simultaneously in a pABI-population. However, an intervention targeting individualized, everyday rehabilitation needs in a home setting has been carried out for the adult TBI-population in the US [34] and is currently ongoing in Norway [35]. This intervention defined individualized rehabilitation goals in collaboration with the patient and family member when available and worked on strategies to reach these goals over a period of four months. To meet the demands of holistic, person-centered and goal-oriented interventions for the pABI-population, this adult intervention was adapted to the pediatric population. In the Child in Context Intervention (CICI), the family and the child’s school are included, and collaboration is established with the Norwegian Service for Special Needs Education (“Statped”). The intervention has largely been adapted to a telerehabilitation format to facilitate access to rehabilitation services, and to ease participation for families and schools.

The Medical Research Council [36] has pointed out the need to carry out feasibility studies as part of the development of complex interventions. This is especially important as the complexity of such studies is influenced not only by the intervention components, but also by contextual factors, in addition to the interactional processes occurring between the intervention components and the context [37]. In pediatric rehabilitation, one also needs to consider a range of transactional processes: between child and parent; between the child, family and school environment; and in the community and with peers, as children rely on their environment even more than adults [38]. In the CICI study, the complex interactions that may take place between children, parents, schools and the intervention components is hard to foresee before the intervention has been tried out. Some key uncertainties include whether the complex logistics of the study are feasible, whether the families are able to maintain participation in an intensive intervention (seven sessions over 4–5 months), whether recruitment is feasible, whether it will be possible to maintain a sustainable working alliance with both children and parents through the telerehabilitation format and whether the goal-oriented approach is feasible when working with an entire family that may not have the same priorities or needs. In addition, as there have been no previous studies carried out including schools in a similar rehabilitation program, it is also of importance to assess the feasibility of this intervention component. Conducting a feasibility study prior to the future definitive RCT is therefore principal in order to evaluate the intervention manual, identify possible obstacles and be able to adjust the protocol before the RCT.

### Objectives

The objectives of the CICI feasibility study were to assess the feasibility of (1) the recruitment procedures and (2) the contents and structure of the intervention, including the feasibility of treatment delivery through a videoconference solution. We also wished to evaluate the acceptability of (3) the intervention for children, parents and therapists, (4) the baseline (T1) and outcome (T2) assessment methods and finally (5) the quality of treatment delivery. In order to inform the final decisions on outcome measures in the RCT, considering preliminary indications regarding the usefulness of outcome measures was also addressed.

## 2. Materials and Methods

This article adheres to the CONSORT extension for the Pilot and Feasibility Trials Checklist [39].

### 2.1. Trial Design

This feasibility trial applied a one-group pre−post design, with a baseline (T1) and a follow-up assessment immediately after the intervention period of 4-5 months (T2). The future definitive RCT will include a two-group RCT-design. For the RCT, outcome assessments will also be performed about 9 months after baseline (T3). The T3-assessment was not included in the feasibility trial to minimize time expenditure as T3 largely matches T2.

### 2.2. Participants and Recruitment Procedures

The inclusion criteria were (1) school-aged children (6–16 years at inclusion) with a radiologically verified diagnosis of ABI, or loss of consciousness post-insult and verified neurological symptoms in cases where radiology could not be administered; (2) time since insult at least 1 year; (3) self- or parent-reported persistent ABI-related cognitive, emotional or behavioral challenges influencing participation in everyday life related to family, friends, school or local community, assessed through a telephone interview; (4) children attending school regularly; and (5) the family is able to participate actively in a goal-oriented study for the next 4–5 months.

Exclusion criteria were (1) severe pre- or comorbid neurological or neuropsychological disorders that would confound assessment and/or outcome measurements; (2) children with brain tumors in active treatment or at great risk of relapse; (3) children with severe psychiatric illness or with injuries so severe that they were currently in institutionalized care; (4) parental severe psychiatric illness, drug abuse or indications of a history of or risk of domestic violence; and (5) not fluent in Norwegian language, although exceptions could be made for English-speaking parents who understand and read Norwegian.

Participants were identified from the medical charts at Sunnaas Rehabilitation Hospital. The families were invited to participate through written age-appropriate information for the children and information to parents. Parents and teenagers from 16 years of age provided written informed consent. A scripted telephone interview was used to screen for inclusion and exclusion criteria and willingness to participate. Eligible and consenting families performed a baseline screening (T1) at Sunnaas Rehabilitation Hospital.

After inclusion, the children’s teachers and principals at their schools received written information about the study and consent form for the teachers. They were thereafter contacted by telephone by the CICI Statped collaborator.

### 2.3. Assessments

The outcome measures to be evaluated in the feasibility study are listed in Table 1. Neuropsychological assessment was included at baseline to provide descriptive data regarding the children with ABI and to inform the goal-setting process. Primary outcomes were post-concussive symptom burden (HBI), parenting self-efficacy (TOPSE) and quality of life (PedsQL). Questionnaires were completed at home and returned by mail.

As part of the baseline assessments, the families were asked to name the current three most challenging areas related to the child’s brain injury. These were rated on a 5-point Likert scale according to how troublesome they were perceived in everyday life. Parents agreed on three areas but scaled them separately. The child was similarly asked to name and rate his or her own three most troublesome pABI-related problem areas.

### 2.4. Intervention

A detailed intervention manual was developed. It was based on the manual developed for an adult population by Borgen and colleagues [35] and adapted to the pediatric and family context by experienced rehabilitation therapists (authors M.L., S.L.H., I.M.H.B., M.V.F., N. R.-B., I.L.H.). In addition to descriptions of the content of each session, the manual contained general information on the study’s rationale and aims as well as detailed guidance on how to establish SMART-goals and strategies, and references to evidence-based strategies and “tools” to handle different challenges. The manual also included detailed guidelines on how to manage therapy and communication through videoconference.

The intervention included seven family sessions, four school-meetings interspersed with the family sessions (starting after the first family-session) and a one-day parent group seminar (occurring about halfway through the family sessions for all participants) over the treatment period of four to five months. Figure 1 gives an overview of the intervention components. The family sessions and the school sessions were delivered through an encrypted videoconferencing solution. The feasibility study was performed from August to December 2020. See Adaptations due to COVID-19 below.

Three therapists, two clinical neuropsychologists and one experienced pediatric nurse (authors N.R.-B., I.L.H. and E.J.S., respectively) with extensive rehabilitation experience delivered the intervention. Therapists received training from the therapists of the adult community-based TBI-study [35,53] concerning the goal-oriented approach and strategies related to different commonly reported ABI-related problem areas. This was discussed in repeated meetings throughout the feasibility study. Therapists delivered nearly all the family sessions in pairs to ensure adherence to the study manual and a uniform delivery across therapists. The school sessions were performed by an experienced special education counselor from Statped (author A.P.S.-S.) in close collaboration with the three therapists. The therapists conducted the parent group seminar. Weekly meetings between the therapists, the Statped counselor and the study Principal Investigator (author M.L.) were held to discuss ongoing therapies and to ensure protocol adherence. The therapists also received training in the technical solutions of telerehabilitation as well as education on important therapeutic aspects of using this format, by a researcher with relevant experience (author H.P.Ø.), and the telerehabilitation team at Sunnaas Rehabilitation Hospital.

### 2.5. Content

#### 2.5.1. Family Sessions

SMART-goals (Specific, Measurable, Achievable, Realistic/Relevant and Timed) [54] were established in collaboration between the family and the therapist in videoconference family sessions. Goal Attainment Scaling (GAS) [55] was established for each goal and recorded in the last session. To increase motivation and comprehensibility for the children, the GAS scaling was set from 1 to 5 instead of using the traditional scaling from −2 to +2, with the preferred starting point being defined as level 2 (equivalent to −1 in the standard GAS). However, in some cases where the desired behavior or action was not present at all at the points of goal-setting, the starting point was 1. A visual presentation of the goal and GAS-scaling in the form of a staircase with five steps was used to support understanding (see Figure 2 for an example). Once a goal was identified, the therapist, parent, and child identified specific strategies designed to achieve and implement the goal. The children collaborated in the goal-setting process according to their cognitive abilities and age. Strategies were based on the available evidence-based recommendations for the pediatric population [15,17,56,57,58,59,60] in addition to recommendations for the adult population with age-appropriate adaptations [61]. Working on and modifying strategies when needed was a main focus throughout the intervention.

Every family received a psychoeducational booklet developed for the CICI study. It was validated by the senior researchers in the project and a user consultant. The handbook consists of 12 short chapters about common challenging areas for families after a pediatric ABI, such as common brain injury symptoms, fatigue, communication in the family, stress management and psychological symptoms in children and their parents, as well as identity issues after pABI. The booklet was used primarily with the parents in relation to goals and strategies set by each family according to how comfortable and interested the parents were in reading such information.

Daily use of strategies was emphasized throughout the intervention to ease transfer to daily life activities for the families.

#### 2.5.2. School Involvement

As the children’s challenges often disrupt their educational or social settings in school, we established school-related strategies related to the families’ goals. The Statped special education counselor visited the schools for observations of the child and the school context. The schools, including the child’s main teacher, were invited to participate in four meetings with the research team. School strategies were established through collaboration between the CICI team, the family and the schools to ensure that the strategies were feasible and adapted to the schools’ environments and resources.

#### 2.5.3. Parent Group Seminar

A one-day interactive parent group seminar focusing on family functioning and parenting challenges was held in accordance with recommendations to include caregivers in pediatric rehabilitation [62]. Topics included: parents’ experiences with SMART-goals, family communication patterns, changes in family dynamics after pABI, how to care for siblings, emotional reactions in the family after pABI (e.g., guilt, grief and embarrassment), and self-care.

### 2.6. Telerehabilitation Delivery

A web-based encrypted videoconference solution was provided by the Norwegian Health Net (join.nhn.no, accessed on 1 February 2020), delivered by Pexip. The solution was risk assessed and approved for clinical use by Sunnaas Rehabilitation Hospital’s Chief information security officer. The participants used their own computers or tablets with integrated camera, and the therapists used their work computers and external microphone speakers to enhance communication. Participants could borrow computers and microphone speakers from the project if they did not have suitable equipment. An IT consultant was available during the sessions in case of technical challenges. Guidelines for therapy through videoconference were developed and conveyed to the participants with recommendations regarding how to create a secure environment for a therapeutic conversation and how to enhance communication through videoconferencing (e.g., one speaker at a time, mute when several participants are joining the same conference, give signs to signal that you want to speak).

### 2.7. Adaptations Due to COVID-19

Due to the COVID-19 pandemic, there were two recruitment periods. Five families were recruited in January and February 2020. Four of these started the intervention program and had completed maximum two sessions per family, which was planned to include two home visits, in addition to the videoconference sessions and the school meetings with physical attendance. However, the study was put on hold in March 2020 due to a national Norwegian lockdown. During this time, one family that had only completed the baseline assessments withdrew. The remaining families received monthly phone calls with information on the study status. One family received psychological support approximately every other week due to high distress levels.

The study re-opened in August 2020, with some adaptations. All family sessions and school sessions were conducted through videoconference (in the original protocol at least the first and last sessions were planned as home visits), and two planned parent groups were reduced to one. Two more families were recruited. All six families started with new baseline questionnaires, interviews and a definition of the main pABI-related problem areas. Children in the two newly recruited families underwent the neuropsychological screening at baseline, but this was not repeated for the four children who were enrolled in January and February.

### 2.8. Procedures of Feasibility Evaluation

To evaluate the study feasibility, distinct objectives were operationalized as shown in Table 2. A more detailed description can be found in the Clinical Trials registration (NCT04186182). The custom-tailored Acceptability Scale was rated on a 5-point scale by children (21 items), parents (40 items) and therapists (33 items): Completely disagree (0), Agree a little (1), Agree moderately (2), Agree (3) and Completely agree (4).

Detailed study-specific checklists were developed in concordance with the detailed descriptions in the manual regarding the content of each session and were used to monitor protocol adherence.

## 3. Results

For protection of privacy, the families are presented in variable order throughout the results section.

### 3.1. Participants

The feasibility trial was carried out with six families, corresponding to 9.4% of the planned total sample size of the RCT (64 families after attrition). This is adequate according to the recommendations for optimal sample size in clinical pilot studies [63,64].

The children were three girls and three boys between 11 and 16 years old at baseline (median 13 years). Time since injury ranged between one and 13 years (median 5.5 years). The injuries were TBI (2), anoxia (2) and brain hemorrhage (2). The mother and father of each child participated, constituting 12 parents in total. All children had siblings and all parents lived together. The majority of the parents had completed 14–16 years of education (seven parents), three parents had 17 years or more and two had 11–13 years of education. Eight parents worked fulltime, while one couple received 50% compensational social support from governmental welfare systems related to their child’s problems due to brain injury. Two parents were on 50 and 100% sick leave. All schools agreed to participate. Four children were in regular schools with some (e.g., structured time-outs, extended time on tests) or no adaptations to their injury-related symptoms; one child attended a private school and had a comprehensive special educational service; and one attended a special educational class. The neuropsychological screening indicated that the range of cognitive functioning overall varied between typical for their age and impaired. See Table 3 for details.

#### Identification of Main pABI-Related Problem Areas

The three most challenging areas related to the child’s brain injury are shown in Table 3. The most commonly identified pABI problem area was fatigue, reported by three parents and three children. In addition to reporting the child’s symptoms as challenging, the parents also reported problem areas related to parenting, worries and communication with the health care system. Overall, it was difficult for some of the children to report their most challenging areas. The therapists put initial effort into getting to know and build trust with the child. The phrasing of the question was adapted to the child’s developmental level and cognitive functioning. When needed, the child was also reminded of the troubles he/she had reported in the questionnaires. Some children still had difficulties with this task, probably due to cognitive deficits such as an underdeveloped ability to generalize and to maintain a meta-perspective on their own level of functioning. Moreover, some children expressed that they did not wish to talk about their difficulties. For ethical reasons, therapists did not push children to define problem areas when they were clearly struggling with the task. The children thus reported fewer challenging areas than their parents. The parents had no trouble reporting three challenging areas in their everyday life related to the child’s injury. The parents of each child agreed on three areas, although they sometimes initially had different opinions on what to choose. The parents often had different opinions regarding how challenging the areas were and therefore scaled them separately.

### 3.2. Objective 1: Recruitment Procedures

Seventeen families were screened for inclusion and twelve were deemed eligible. Of these, seven families (60%) were willing to participate (see Figure 3). The reasons for declining participation were not experiencing challenges that the family currently needed help with (*n* = 2) and not having enough time (*n* = 3). The recruitment rate was deemed highly feasible according to the predefined criteria, as we had set the a priori level of highly feasible to 30%, and we included 60% of the families we approached. Given that the existing literature indicates that at least 30% of these families experience unmet needs [9,11,65], we would expect an inclusion rate in the same range or higher when recruiting from a rehabilitation hospital. Time spent on recruitment for each family was also deemed feasible (less than 3 h per family). None were excluded after baseline, indicating that the screening process was satisfactory. One family withdrew before starting the intervention during the COVID-19 lockdown, due to having second thoughts on whether the intervention would help their child. All families that started the intervention completed it. In total, the recruitment procedures were highly feasible.

### 3.3. Objective 2: Contents and Structure of the Intervention

#### 3.3.1. Attendance

Members from all six families completed the family sessions (100% family attendance), and all families were represented in the parent seminar. Due to illness, one parent neither completed the second half of the intervention nor the T2 assessment. However, the child and the other parent completed as planned. All four school meetings were conducted for all participants (100% school attendance), and four families participated in at least one school meeting. The rest of the meetings that parents did not attend were forgotten by the parents. On average, the family sessions lasted 98 min including breaks. The children attended parts of all the sessions, according to what topic was being discussed and the child’s concentration and willingness to participate. All families received extra telephone follow-ups related to their goals and strategies in addition to the procedures described in the manual (with a total duration of 20 to 150 min per family). For two families, telephone contact was also made to other collaborators: physiotherapist (20 min), school nurse (20 min) and the special educational service (45 min) for one family, and two phone calls to the child’s assistant for the other family (40 min). Overall, the high attendance rates indicate that the intervention implementation was feasible.

#### 3.3.2. Evaluation of the SMART-Goal and GAS Approach

All families set SMART-goals that were related to some or all of the main problem areas they reported at baseline (Table 3). For two families, new areas to work on became evident during the intervention. Five families defined three goals and one family defined five, providing a total of 20 goals. Of these, six goals had their starting point at GAS 1, whereas the rest started at GAS 2. The most frequent topic was fatigue, which was the focus of at least one goal in five families. Increased independence in everyday life was a topic for two children, including leaving the house on his/her own in the morning, keeping track of appointments, taking the bus and starting to ride a bike again. Two worked on goals to reduce pain and two had goals regarding social functioning. Two families worked toward parental mental health goals, and two families had goals regarding family communication and learning how to talk about the injury with others. One family aimed to apply a problem-solving technique and one set a goal related to the child’s study skills.

The children’s participation in the goal-setting process varied according to their abilities and motivation. For instance, the youngest child (11 years) would participate in setting the goal and name already existing good strategies to obtain the goal (for instance, rest after school). The child would also participate in discussing new possible strategies to ensure ownership and collaboration (for instance, how the child would be comfortable resting at school). Due to short attention span, strategies related to the parents’ actions would be discussed when the child took a break (for instance, encouraging the child to name the level of experienced fatigue three times a day on an “energy-scale” and taking notes of the different activities the child had endured that day). In contrast, one of the teenagers was cognitively able to participate in larger parts of the sessions, the goal-setting processes and discussions on strategies. During the family sessions, all children were encouraged to share their experiences in working with the strategies since the last session and to state their opinions on whether the strategies were helpful and feasible for them. The strategies were adapted and/or new strategies were established as needed. To a large extent, the strategies were external, which means adapting the environment or facilitating the establishment of new skills. One example is parents who trained their child to use a smart-watch by using principles of errorless learning [66], where the parents gradually offered less help as the child gained confidence and skills. Implementation of the strategies in everyday life was highlighted throughout the intervention by basing the strategies on the individual family’s everyday life routines and resources and through encouraging daily use of the strategies.

The families attained all their goals but one (in family B). For 14 of the 20 goals, goal achievement was beyond the expected level on the GAS. Figure 4 shows goal attainment scaling per goal for each family. None of the goals showed negative GAS change.

#### 3.3.3. Responses to the SMART-Goal Approach

Both parents and children perceived the SMART-goals as highly relevant, with all but one score ranging from 3 (agree) to 4 (completely agree) on the corresponding item on the Acceptability Scale. See Table 4 for the individual ratings. All children confirmed the importance of the goals and were pleased to achieve the skill, but some of them found working on the skills and spending time in meetings instead of being at school or with friends tiresome. The parents reported that the strategies to achieve the goals had helped their children, with four families responding with 4 (completely agree) on this item and two families scoring 3 (agree). Overall, the SMART-goal and GAS-methodology was deemed highly feasible.

#### 3.3.4. The Use of Videoconferences in Treatment Delivery

Overall, the technical solutions worked very well. All families but one had excellent internet connection, and every family owned equipment suitable for videoconferences (PC or tablet). Support was provided to the family with slow internet connection, and solutions were found to enhance the quality of the videoconferences. External microphone speakers were sent to the families to optimize the sound. There were only a few incidents of needing to restart the equipment (less than one session per family) throughout all of the 40 videoconferences. According to the predefined criteria, the technical solutions were highly feasible.

The satisfaction with the use of videoconference in the intervention (see Table 4) was rated as high by both parents and children (median score 3 for both). The therapists rated the use of videoconferences in the intervention as good overall (median 3) and also experienced it as highly feasible to set goals and strategies and to implement the intervention with the family (median score 4). However, the therapists rated it as challenging to maintain good communication with the children through videoconference, with most ratings at the lower end of the scale on the question framed “communication was good with the child”. The therapists’ ratings varied from 0 to 4, with the lowest rating being in regard to a child with very severe cognitive deficit. Overall, the use of videoconference was evaluated as an acceptable approach for treatment delivery, but with a special focus on involving the children.

### 3.4. Objective 3: Acceptability for the Children, Parents and Therapists

#### 3.4.1. Working Alliance in the Intervention

The parents’ ratings of working alliance were high (median score of 4, all scores either 3 or 4), whereas the children showed more variation, but with median scores at the high end of the scale (median 3, range 0.5 to 4). See Table 4. Two of the children rated the working alliance as low (1.5 and 0.5). These were one teenager who expressed that he in general did not want to focus on the brain injury, and also that he was not happy that the therapists and his parents talked about his challenges in his absence. The younger child was reluctant to participate in the last sessions, although the goals were achieved.

The therapists’ ratings of working alliance were high with regards to feeling welcome to contact the family and the general tone of communication (median 5 for both), but two questions concerning relation and communication with the child had more variable responses, similar to the children’s own ratings (median 3, range 1–5). The therapists rated the collaboration with the schools as good (median 4.5, range 3–5).

#### 3.4.2. Usefulness of the Intervention

The parents rated the intervention as highly useful (median 4, range 3 to 4). The children’s ratings showed large variations also for usefulness, ranging from 1 to 4 with a median of 3. The variability between the children was similar to the responses on the questions related to working alliance (see Table 4), with yet another teenager rating the usefulness as low. The parents rated the CICI handbook as very useful (median score 4). Some parents reported that they read the entire booklet several times during the intervention, whereas others mainly used the booklet during psychoeducation in sessions. The parent group seminar was rated as overall very useful (median score 5).

The therapists rated their experience of the usefulness of the intervention for the families as high (median 5, range 2–5). Only 1/77 responses was a 2, which was due to the fact that the therapist considered that a child with very severe cognitive deficit had not gained an increased understanding of his/her symptoms due to the intervention. The remaining items were scored within a range of 3–5.

Parents of two of the children who gave low ratings of working alliance and usefulness specifically commented that the child was tired and in a bad mood when filling out the Acceptability Scale, which in their opinion had influenced response validity. Overall, working alliance and usefulness were judged as highly feasible according to the predefined criteria for both parents and children, although with notable variation in responses among and regarding the children.

### 3.5. Objective 4: Methods of Assessment at Baseline and T2

#### 3.5.1. Burden of Assessments

Regarding the burden of assessments, all children reported that they understood the questionnaires and all but one reported that it was good to be able to report on how they were doing through the questionnaires. However, three children were very fatigued from the assessments, as reported by both children and parents. In addition, three children experienced the neuropsychological assessment as very burdensome. Four of the 11 parents reported that their child had trouble understanding the questionnaires. Two parents reported that there were too many questionnaires, while the remaining parents reported that they did not find the number of questionnaires too burdensome. All parents but one agreed that the topics of the questionnaires were relevant.

The duration of the baseline assessment was on average more than 4 h per family, equivalent to “not feasible” on our predetermined criteria. In addition, we experienced challenges in collecting questionnaires from both parents and children at T2. In summary, the assessments were considered too lengthy and too much of a burden especially for children, but also for the parents.

#### 3.5.2. Outcome Measures

The feasibility of the outcome measures was assessed with the aim of informing the final decisions regarding primary and secondary outcomes in the RCT. The results of the questionnaires were explored, but no group-average-based statistical analyses of change were performed. The individual scores for selected questionnaires are presented. Group mean and median scores are not presented as the main aim of a feasibility trial is to assess the intervention protocol, not the outcomes. Furthermore, the small sample does not render statistical group analysis useful.

When inspecting the planned primary outcomes (post-concussive symptom burden (HBI)), parenting self-efficacy (TOPSE) and the child’s quality of life (PedsQL), there appeared to be low correspondence between parent- and child-ratings. The results of the HBI displayed a lower symptom burden post intervention for all participants on the somatic subscale, and for 10 of 17 respondents on the cognitive subscale (see Figure 5). The TOPSE (parenting self-efficacy) showed improvement (higher scores) for eight of the eleven responding parents after the intervention, with the largest positive changes in fathers (see Figure 6). The PedsQL, however, showed a less consistent pattern of change post intervention, with improvements reported by three of six children and eight of the eleven parents (see Figure 7). All parents but one mother and one father rated their family needs (FNQ-P) as met to a larger degree after the intervention. Emotional distress in parents varied, with more depressive symptoms at T2 for six of ten parents (PHQ-9), and more anxiety symptoms in four of nine parents (GAD-7). However, only one parent scored above the clinical cut-off post intervention, and one mother who reported moderate symptoms pre intervention was below the cut-off post intervention. The emotional symptom burden was thus in the low range for all parents but one at T2.

### 3.6. Objective 5: Quality of Treatment Delivery

In total, there were only small deviations from the study protocol, with a mean adherence of 95.6%. The 4.4% deviations resulted mainly from the fact that the psychoeducational CICI handbook was used less than expected in the individual sessions. The quality of treatment delivery was deemed highly feasible according to the predefined criteria.

### 3.7. Harms

There were no reported harms or unintended effects.

## 4. Discussion

Although the intervention was found to be feasible overall, valuable information was obtained on issues that needed to be considered before the future definitive RCT.

### 4.1. Contents and Structure of the Intervention

Recruitment rates were high in this feasibility study, as 60% of the eligible families were willing to participate. Of the declining families, two did not report challenges which they needed help with, and three did not have the time to participate in an extensive rehabilitation program at this point. The participating families were recruited from Sunnaas Rehabilitation Hospital, where children with specialized rehabilitation needs are referred to after acute care. In the future definitive RCT, patients will also be recruited from the acute hospital of the South-Eastern Health Region and from the national special education service, providing a population with a broader spectrum of severity and possibly different long-term needs. The RCT-recruitment will thus include less severe injuries. Furthermore, the relation that the recruited families had to Sunnaas Rehabilitation Hospital may have influenced their willingness to participate in this study. The fact that we recruited participants from a rehabilitation hospital may indicate a selection bias toward participants with severe injuries and therefore a high level of unmet needs. The final inclusion rate in the future RCT remains to be established but may be expected to be somewhat lower.

Attendance rates were very high. However, the attendance of parents in the school meetings was lower than expected, which was interpreted as too many meetings during the intervention. It was, however, not crucial that the parents attended all school meetings. The CICI therapists made sure that important information was shared between the families and schools, and emphasis was put on establishing means of communication before the intervention ended. The high attendance rate of the schools showed that it was feasible to include schools in the intervention. In the future definitive RCT, parents will be offered to attend school meetings to the extent that they find useful and manageable. Contact with local health care providers was established for two of the six participants. Interestingly, most of the participants did not receive help from local health care providers, confirming the high incidence of unmet needs in areas of, for instance, fatigue, cognitive rehabilitation and issues related to increased independence in everyday life. In this respect, the CICI provided services that the families would not have received elsewhere.

Defining the three main pABI-related problem areas of daily life worked well for the parents, but it became apparent that parents of the same child did not always experience the same problem areas or experience the problems as equally challenging. In addition to being able to scale the pABI-related problem areas separately, parents will also have the opportunity to define separate areas in the future definitive RCT, to avoid important areas being overlooked. Some of the children struggled with this task, and clinical consideration will be taken in the future RCT, as was done in the feasibility trial, by accepting a lower number of problem areas from the children when necessary.

Overall, the satisfaction with the SMART-goal approach was high in parents and children. Interestingly, the SMART-goals were obtained and perceived as useful also for the children who responded with low ratings of working alliance and usefulness of the intervention. This finding may reflect that children have poorer abilities of abstract thinking and generalizing than adults. It may reflect a common challenge in all therapy with children: children rarely seek help by themselves, they have a less developed insight into their challenges, they are less motivated to change, and they often have a different understanding of their problems and how to solve them than their parents [67]. This influences children’s motivation to take part in treatment. In addition, children with brain injuries have varying degrees of awareness of their deficits, further adding to reluctance to participate in treatment.

Goal attainment was high. Although there was some variation in goal attainment (Figure 4), all but one of 20 goals reached at least the expected level of achievement. The variation in goal attainment might depend on the complexity of the goal. High goal attainment (highest level of GAS) was achieved for less complex skills such as learning to ride a bike, whereas goals related to more complex skills, e.g., communication and mental health, showed progress as expected. The high goal attainment showed that it was possible to achieve positive change in symptom areas that are common after pABI, such as fatigue, independence in everyday life, pain and problem solving. The SMART-goal approach was thus a feasible and appreciated method for working with a broad range of problems. Whether the intervention as a whole will have a significant effect remains to be established in the definitive RCT.

### 4.2. Acceptability

The families responded well to the use of videoconference in treatment delivery, and the technical solutions were satisfactory. The therapists found that using videoconferences worked surprisingly well for building trust and for treatment with the parents, but it was more challenging to establish a high-quality communication with the child. In line with this, the perceived working alliance and usefulness of the intervention was higher for parents than for the children, but with large variation among the participating children. Some of the children tended to disappear from the video meetings when they lost concentration. Some of them did not want to talk much in the sessions, which made it difficult for the therapists to engage them. This was experienced as especially challenging in communication with the teenagers, who seemed more reluctant to focus on the brain injury than the younger participants. Unfortunately, this feasibility study did not succeed in recruiting the youngest children (from age 6 to 11).

Building a relationship with a child is facilitated by establishing joint attention and engaging in joint activities, which is challenging in videoconferences. Maintaining the child’s attention is often facilitated by eye contact and by the therapist’s ability to adjust conversational strategies to the child’s needs, which may be more difficult to achieve through videoconferences. In addition, building alliances with children in therapy is complicated by the fact that therapists also need to establish an alliance and negotiate goals with caregivers as well [68]. Thus, general aspects concerning the treatment of children were seen that may not have been directly related to the intervention being videoconference-based, although videoconference may have amplified them. On a positive note, research on treatment effects and alliance in therapy with children and families has found that the alliance with parents influences treatment outcomes more than the alliance with the child [69]. The therapists in this intervention rated communication with the parents through videoconference as good. However, they rated it as more challenging to maintain good communication with the children, even with the children who rated their own satisfaction with the intervention as high. The fact that therapists may rate the satisfaction with the telerehabilitation lower than the participants has been found in other studies [70] and may be influenced by the complex therapeutic tasks. In a telerehabilitation environment, therapists face several tasks simultaneously: preserving therapeutic alliance, delivering therapy and dealing with technical difficulties, which demands multitasking beyond face-to-face delivery. Participants, however, tend to display a higher technology failure tolerance than the therapists [71]. These factors may have influenced the therapists’ experience of the telerehabilitation communication with the children, where expectations from the experienced therapists were high beforehand.

For the future definitive RCT, some of the intervention material will be further developed to engage the child in conversations and to establish a sense of ownership to the intervention. Although most children and parents reported gains through the intervention and appreciated the accessibility that video sessions provided, a videoconference-based-intervention may be particularly challenging for some children. The children’s participation in the sessions and ability to generalize and reflect on their experience will necessarily vary according to factors such as the child’s age, state of mind, cognitive difficulties, level of fatigue and personality, as well as the child’s relationship to and interaction with their parents and the therapeutic alliance. In addition, the children’s state of mind at the time of completing the Acceptability Scale seemed to influence how they responded, possibly influencing the validity of their responses. The intervention was conducted during the COVID-19 pandemic, and the children’s ratings may also have been influenced by both frustrations related to the lack of normal activities in their lives, and perhaps a low motivation for videoconferenced activities at a time where school was mostly conducted through this medium for the teenagers.

Due to the COVID-19 pandemic, conducting an intervention through videoconference enabled the provision of health care services that would not otherwise have been possible. In Norway, most families have grown accustomed to using videoconference as a medium of communication, as both school and work have been carried out through digital media during lockdown for a large part of the population. As such, the pandemic has changed the prerequisites for a telerehabilitation intervention, making it more available.

### 4.3. Methods of Assessment at Baseline and Post Treatment

The baseline assessment protocol was too lengthy and burdensome for children and parents. Adaptations will thus be made for the future RCT. Firstly, the neuropsychological screening on baseline will be reduced to only two subtests of abstract thinking (Matrix and Similarities from WISC-V). The reduction in neuropsychological measures was deemed appropriate as the main focus of the intervention is on everyday challenges, regardless of cognitive profile. Secondly, reducing the number of questionnaires for children and parents was also necessary. As CICI is an individualized intervention, it is challenging to define one common outcome at the group level. After careful considerations, we decided to include outcome measures that target areas that are commonly experienced as challenging after pABI [9,14], and which we also expect will be targeted in the intervention. Furthermore, we wished to include broader domains such as quality of life and participation. Given the family focus, it was important to also include measures that would capture parent factors such as parent mental health and parenting self-efficacy, as well as family function. The feasibility study provided important information on the selected assessment methods which, together with a thorough literature review, was used to inform the final decisions on assessment and outcome methods in the future definitive RCT. Three questionnaires (CASP, PSS and SDQ) were excluded as they were judged to have significant overlap with other questionnaires, appeared to not be very sensitive to change, and/or were judged to contribute with less important information for the study purpose. This feasibility trial also aided in the determination of what should be primary outcome measures in the future definitive RCT. Due to correction of the alpha level according to multiple primary outcomes, a maximum of two primary outcomes was decided to ensure adequate statistical power with a feasible sample size. The large variability and possible low validity in the children’s responses led to the decision to use parent ratings as primary outcomes, which is common in family interventions and interventions including children with brain injury [72,73,74,75]. To be able to capture changes in symptom severity in the child as well as important parent factors [49], changes in parent-reported brain injury symptom severity (HBI) and parenting self-efficacy (TOPSE) were thus chosen as primary outcome measures. The final CICI protocol with all changes resulting from the feasibility study is described in detail in a published CICI protocol article [76].

Regarding the questionnaire results, the positive feedback on the Acceptability Scale appeared to be captured in some of the measures, such as reduced brain injury symptoms reported by parents (HBI), lower levels of executive deficit (BRIEF), improved quality of life (PedsQL), higher parenting self-efficacy (TOPSE) and fewer unmet family needs (FNQ-p). Although some parents reported more emotional symptoms after the intervention, only one had symptoms equivalent to moderate depression. The elevated symptoms might reflect a more accurate rating of emotional state at T2, as the therapeutic alliance results in more openness from the parents. On the other hand, parents face long-term challenges that are likely to not be fully overcome in a 4–5-month intervention. Particular interest should be devoted to this issue in the future RCT, as we should be cautious about the risk of parents feeling overwhelmed at the prospect of again being left to deal with their problems on their own.

### 4.4. Limitations

The low number of participants in this study constitutes a limitation regarding generalizability of the results, especially concerning the results of the outcome measures. However, the main purpose of this study was to assess feasibility and not to investigate statistical effects. The participating children had different types of injuries and a large span in time since injury, which was considered a strength, while the restricted variation in age should be considered a limitation. In addition, the parents’ educational level was high and all parents were married, which reduces the generalizability of the findings.

## 5. Conclusions

The findings from the CICI feasibility study indicate high intervention feasibility and acceptability, particularly for the parents. As the use of external strategies in cognitive rehabilitation of children with ABI tends to be the most reliable approach, it was considered acceptable and to a certain degree expected that the alliance and communication with the parents was superior to that of the children. The focus on SMART-goals was perceived as useful by all participants, and including schools in the intervention proved beneficial. The telerehabilitation format seems acceptable, although some concerns regarding the engagement of children need to be monitored. The number of neuropsychological tests and questionnaires was reduced, and new primary outcome measures were defined as parent-reported brain injury symptom severity (HBI) and enhanced parenting self-efficacy (TOPSE). Except for this, no major adjustments to the protocol were made (see protocol article by Rohrer-Baumgartner et al. (2022) REF).

## Figures and Tables

**Figure 1 jcm-11-02564-f001:**
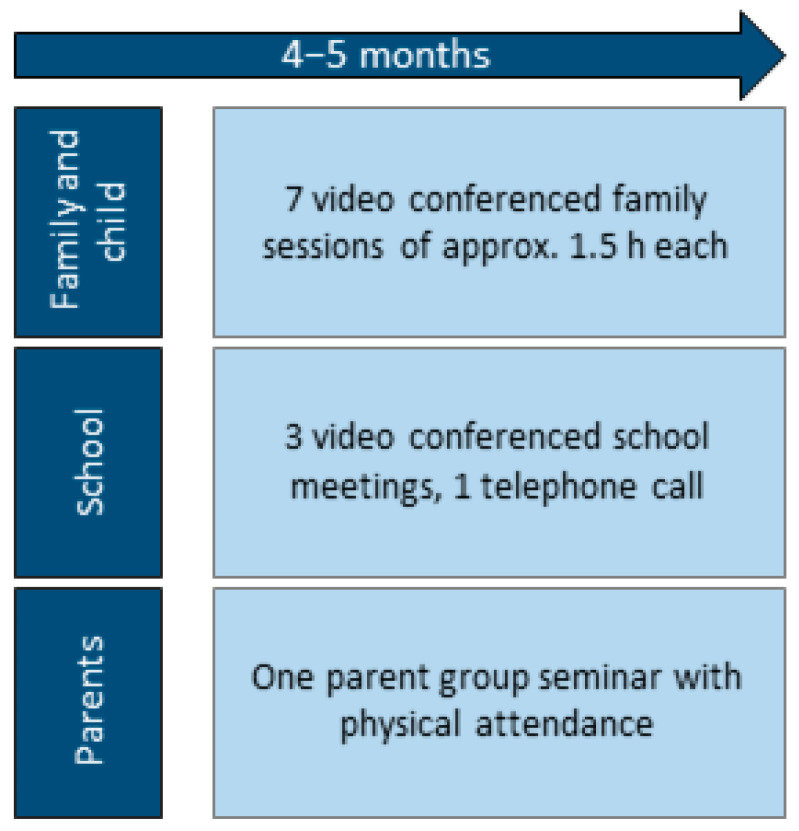
Overview of the intervention.

**Figure 2 jcm-11-02564-f002:**
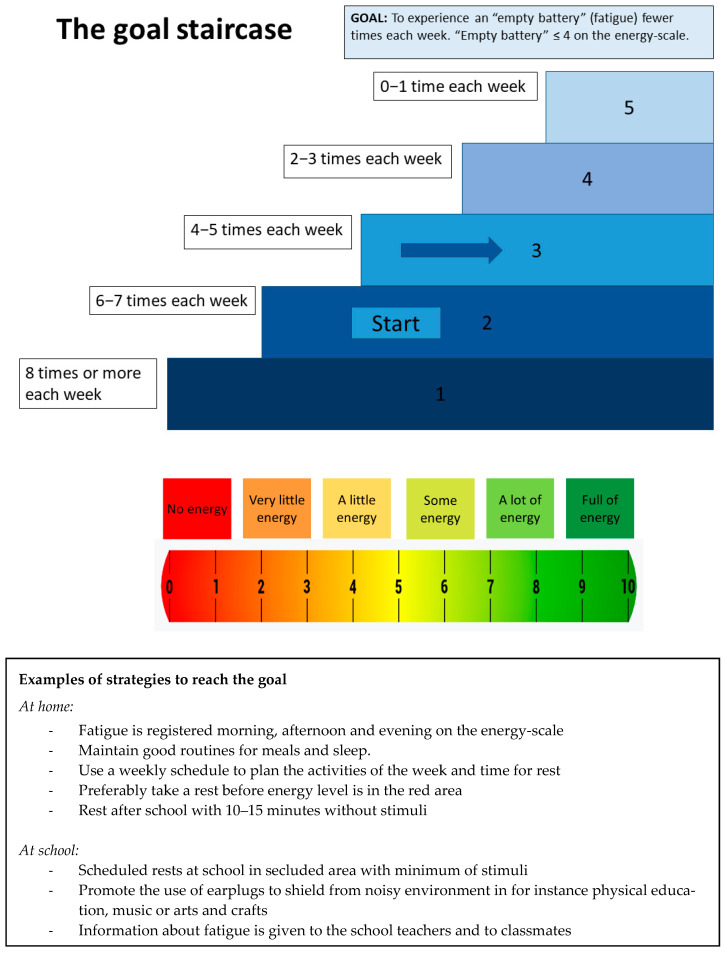
An example of the “The goal staircase”. This example shows a goal related to fatigue and how fatigue was operationalized as energy on a scale. Energy level is measured three times each day. Strategies to obtain the goal are presented in the textbox.

**Figure 3 jcm-11-02564-f003:**
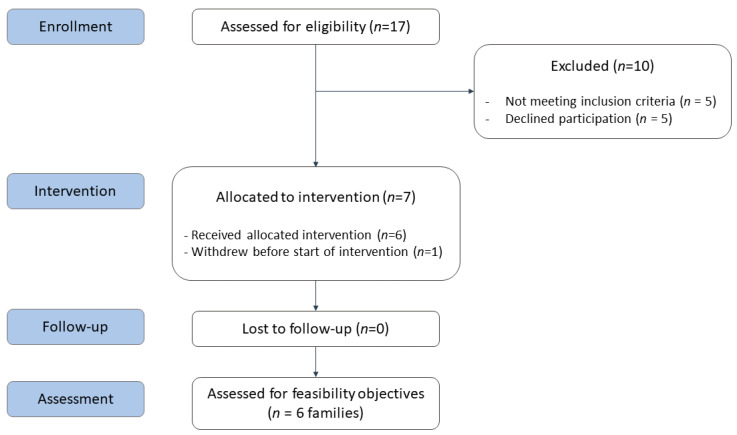
Illustration of the recruitment process.

**Figure 4 jcm-11-02564-f004:**
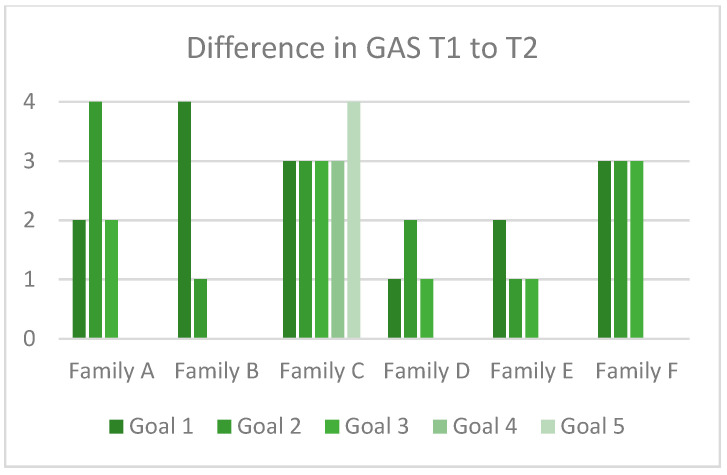
Goal attainment scaling on each goal per family, measured by the GAS change from T1 to T2. A positive number means that the goal was achieved. For family B, one goal had no progress on the GAS and is not visible in the figure.

**Figure 5 jcm-11-02564-f005:**
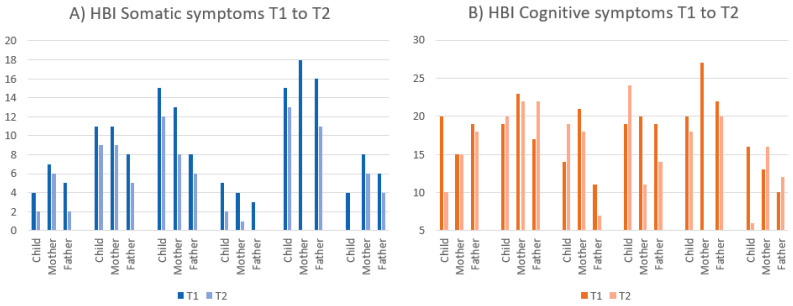
Ratings on the (**A**) somatic and (**B**) cognitive sub-scales of the HBI at T1 and T2 for each family. Lower scores imply lower symptom burden.

**Figure 6 jcm-11-02564-f006:**
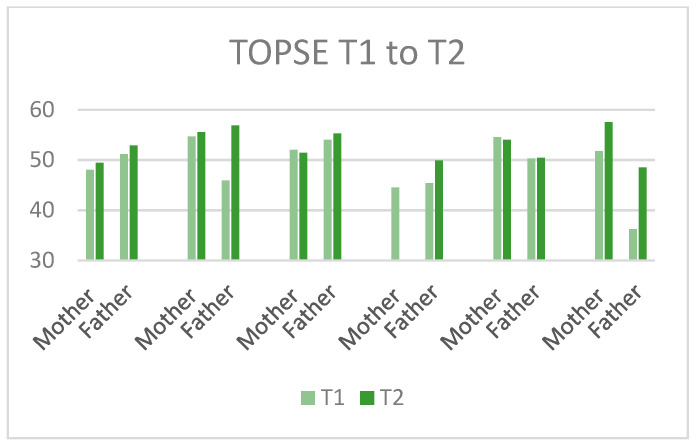
Ratings on the TOPSE each family at T1 and T2. Higher scores imply higher parenting self-efficacy.

**Figure 7 jcm-11-02564-f007:**
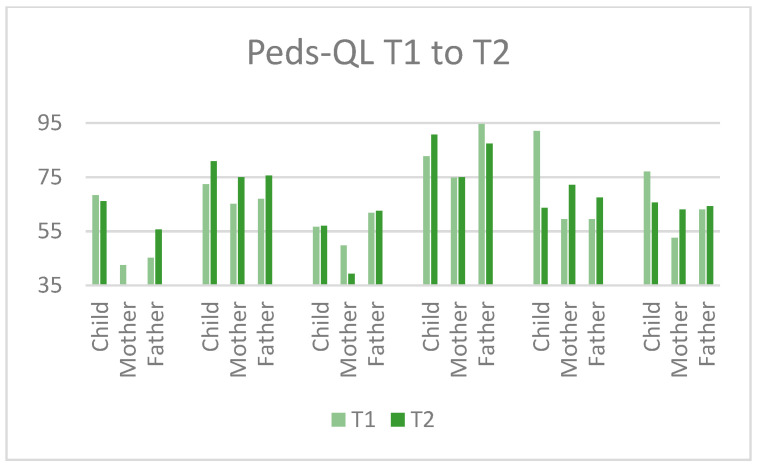
Ratings on the PEDS-QL for each family at T1 and T2. Higher scores imply higher reported quality of life.

**Table 1 jcm-11-02564-t001:** Measures included in the feasibility study.

Assessment Domain	Instrument
* **Neuropsychological Assessment at Baseline Only** *	
Verbal IQ estimate	Similarities from Wechsler Intelligence Scale for Children (WISC-V) [40]
Non-verbal IQ estimate	Matrix reasoning from WISC-V
Auditory attention/verbal working memory	Digit span from WISC-V
Visuomotor processing speed	Coding from WISC-V
Verbal learning and memory	Children’s Auditory Verbal Learning Test-2 (CAVLT-2) [41]
Verbal inhibition	Inhibition from the Developmental Neuropsychological Assessment (NEPSY-II) [42]
Auditory comprehension	Comprehension of instructions from NEPSY-II
* **Questionnaires at Baseline and Post Intervention** *	
Emotional, behavioral and social functioning	Strengths and Difficulties Questionnaire (SDQ) [43]
Participation: home, neighborhood, community	Child and Adolescent Scale of Participation (CASP) [44]
Quality of life	The Pediatric Quality of Life Inventory (Peds-QL) [45]
Post-concussive symptoms after ABI	Health and Behavior Inventory (HBI) [46]
Executive functioning at home and in school	Behavior Rating Inventory of Executive Function-2. ed. (BRIEF-2) [42]
Main pABI-related problem areas of daily life	Likert scale from 0 (Not at all difficult) to 4 (Very difficult)
Family functioning	Family Assessment Device (FAD) [47]
Parent self-perceived stress	Parental Stress Scale (PSS) [48]
Parenting self-efficacy	Tool to measure Parenting Self-Efficacy (TOPSE and Teen TOPSE) [49]
Unmet healthcare needs of the family	Family Needs Questionnaire Pediatric Version (FNQ-P) [50]
Parents’ depression symptoms	Patient Health Questionnaire—9 item (PHQ-9) [51]
Parents’ generalized anxiety symptoms	The General Anxiety Disorder—7 item (GAD-7) [52]
Acceptability of intervention, self-tailored	Acceptability Scale rated on a Likert scale from 0 (Completely disagree) to 4 (Completely agree)

**Table 2 jcm-11-02564-t002:** Objectives of the feasibility evaluation with predefined criteria.

Objective Assessed by	Predefined Criteria
**Objective 1: Recruitment procedures**	
Consent rate.	Highly feasible: More than 30% consent rate Moderately feasible: 15–29% consent rate Not feasible: Less than 15% consent rate
Duration of recruitment processes.	Highly feasible: Less than 3 h per family spent on recruitment Moderately feasible: Between 3 and 5 h Not feasible: More than 5 h
Number of participants excluded at or after the baseline assessment to reach 6 participation families.	Highly feasible: One or no families excluded at or after baseline Moderately feasible: Two families excluded at or after baseline Not feasible: More than two families excluded at or after baseline
Drop-out rate.	Highly feasible: No drop-outs Moderately feasible: One drop-out Not feasible: Two or more drop-outs
**Objective 2: Contents and structure of the intervention**	
Attendance rate.	Measured in % attendance
Feasibility of the SMART-goal approach by feedback from participants on three items on the Acceptability Scale concerning the importance of the goals, and how helpful the strategies were for the child and for the family.	Highly feasible: Median score over 3 (“Agree”) Moderately feasible: Median score between 2 (“Agree moderately”) and 3 (“Agree”) Not feasible: Median score lower than 2
Feasibility of videoconference in treatment delivery as assessed by:	
- One question in the Acceptability Scale concerning the quality of communication through videoconference rated by the children, their parents and the therapists.	Highly feasible: Median score over 3 (“Agree”) Moderately feasible: Median score between 2 (“Agree moderately”) and 3 (“Agree”) Not feasible: Median score lower than 2
- A technical log, where number and type of technical failures were reported by the therapists.	Highly feasible: Restart of equipment in 0–1 session per family Moderately feasible: Restart in 2–3 sessions Not feasible: Restart in more than 4 sessions per family
**Objective 3: Acceptability for the children, parents and therapists**	
Working alliance in the intervention was measured by child and parent ratings on six items concerning the relation with the therapist; the experience of being heard, taken seriously and given information; and whether they would recommend the study to others. In addition, working alliance was rated by the therapists on five items concerning the experienced quality of relationship with the families.	Highly feasible: Median score over 3 (“Agree”) Moderately feasible: Median score between 2 (“Agree moderately”) and 3 (“Agree”) Not feasible: Median score lower than 2
Usefulness of the intervention was rated on six items on the Acceptability Scale for the children and nine items for the parents, concerning the helpfulness of the intervention, the knowledge transfer to other situations and whether one learned something new. In addition, the therapists rated their experience of the usefulness of the intervention for the families on seven items concerning helpfulness of the intervention, importance of the goals, usefulness of the parent group seminar and awareness of and openness toward the child’s challenges.	Highly feasible: Median score over 3 (“Agree”) Moderately feasible: Median score between 2 (“Agree moderately”) and 3 (“Agree”) Not feasible: Median score lower than 2
**Objective 4: Methods of assessment at baseline and T2**	
Burden of assessment was rated on the Acceptability Scale by four children, and parents rated items concerning whether the child was comfortable being tested and expressing his/her symptoms and opinions through the questionnaires, understood the questionnaires, and was fatigued by the assessments. Parents also rated two items concerning the number of questionnaires and the relevance of the topics in the questionnaires.	Highly feasible: Median score over 3 (“Agree”) Moderately feasible: Median score between 2 (“Agree moderately”) and 3 (“Agree”) Not feasible: Median score lower than 2
Duration of the baseline assessment.	Highly feasible: Less than 3 h Moderately feasible: 3 to 4 h Not feasible: More than 4 h
**Objective 5: Quality of treatment delivery**	
Protocol adherence by study-specific checklists monitoring discrepancies between actual intervention delivery and the CICI manual.	Highly feasible: Less than 15% deviation Moderately feasible: 16–25% deviation Not feasible: More than 25% deviation

**Table 3 jcm-11-02564-t003:** Neuropsychological functioning and main pABI-related problem areas.

Family	Neuropsychological Functioning	Parents’ Identified Problem Areas	Child’s Identified Problem Areas
1	Within normal range	Fatigue, emotion regulation, study technique	Fatigue, study skills
2	Impaired memory and verbal reasoning (≤−2 sd). Slightly impaired processing speed, working memory and visual reasoning (≤−1 sd).	Fatigue, cognitive gap to peers, worry for child’s emotional health	Fatigue
3	Executive dysfunction and impaired processing speed (≤−3 sd), impaired working memory (−2 sd), reduced memory functions and verbal reasoning (≤−1.3 sd).	Social challenges, headache, fatigue	Social challenges, headache
4	Reduced working memory (−1.3 sd)	Parenting a child with ABI, child’s social insecurity, pain	Pain, sleep, fatigue
5	Overall, severely reduced neurocognitive functioning with all scores in the impaired range (between −1.3 to −3 sd, with all but 2 tests ≤−2 sd)	Parental exhaustion tied to challenges in getting adequate help for child, child’s social isolation	Losing track in conversations with peers, not able to follow activities and changes in the same tempo as peers
6	Executive dysfunction (≤−2.3 sd), impaired processing speed (−2 sd) and reduced visual reasoning (−1.3 sd)	Social challenges; physical challenges such as balance, coordination and strength; lack of independence in getting around	Getting around independently

**Table 4 jcm-11-02564-t004:** Working alliance, usefulness and evaluation of SMART-goals and strategies, scale from 0 (“Completely disagree”) to 4 (“Completely agree”).

Participant	Relevance of SMART-Goals	Helpful Strategies for the Child ^1^	Satisfaction with Video-conference in Treatment	Working Alliance	Usefulness
Child	4	-	3	4	3.5
Mother	4	4	4	4	4
Father	4	4	4	4	4
Child	3	-	3	1.5	2
Mother	4	3	3	4	3
Father	3	3	3	4	3
Child	4	-	3	0.5	2
Mother	4	4	4	4	4
Father	4	4	3	4	4
Child	2	-	3	3	1
Mother	*	*	*	*	*
Father	3	3	3	4	3.5
Child	4	-	3	4	4
Mother	4	4	3	4	4
Father	4	4	4	4	3.5
Child	3	-	3	3	3
Mother	3	4	3	4	3
Father	3	4	3	4	3

* Indicates missing data. ^1^ Parent rated

## Data Availability

The data presented in this study are not publicly available due to protection of privacy.

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
