# Peer review of "Feasibility and Acceptability of a Complex Telerehabilitation Intervention for Pediatric Acquired Brain Injury: The Child in Context Intervention (CICI)"

_jcm, 2022, doi:10.3390/jcm11092564_

Round 1
Reviewer 1 Report
You have presented a very clear and useful feasibility study. This is particuallry commendable given the complexity of the intervention.
Please refer to attached pdf for suggestions on this manuscript.

Author Response
Thank you for your thorough and interesting review of our manuscript.
Please see the attachment for responses to your comments.
All the best, Ingvil Laberg Holthe

Reviewer 2 Report
Comments to the Authors
This is an interesting article focusing on the feasibility of the Child in Context Intervention (CICI), an innovative individualized, goal-oriented and home-based intervention for children after acquired brain injury and their families.
The article is well-written and clear in its proposals. Methodology is described accurately.
I report below some suggestions to further improve the manuscript quality:
Introduction
The introduction could benefit from presenting a brief summary of the other tools used for remote technology-based metacognitive rehabilitation in young patients with acquired brain injury, considering that some evidence on the topic has already been gathered. The largest number of experiences with this form of rehabilitation has been collected in the USA, but are there some studies also in other countries, particularly in Europe?
Methods
With respect to the intervention, in line 194 it is reported that ‘A detailed intervention manual was developed’. Could the Authors provide more information on such a manual? Who defined the contents? Which was the rationale sustaining the development of the intervention and the content sessions in the actual form?
Table 2: With respect to section Objective 2, in the right column ‘Predefined Criteria’ for the second item the description of outcome measures is reported to be as ‘Same as above’. However, in Objective 3 the same outcome measures’ description has been reported for item 1 and item 2. Please choose the same criterion to report information in the table.
Could the Authors better explain the meaning of the sentence reported in lines 352-353 ‘The recruitment rate was in line with expectations, based on existing literature showing that the level of unmet health care needs in this group is high’? What does existing literature report about unmet health care needs of children with ABI? How this information could justify reasons for declining participation of families contacted for this study?
Table 4: for better clarity, please indicate the number of the participant for which child, mother and father reports are reported. At the moment the distinction between participants is only based on the color (white or grey) of the row.
Discussion
The discuss would benefit from inserting more detailed information on the clinical implications of this study.
Further, it should be described and discussed, as a feasibility data, the percentage of families declining to participate into the study and the reasons associated with this choice. This, in fact, could provide important indications on the usability of the program and on possible selecting bias with respect to the sample included in the study. For the final randomized clinical trial, it could be useful to collect anagraphic and demographic data on families not accepting to partake into the project, as it would provide important knowledge on persons not willing to insert telerehabilitation in the routine care of their children. Is there any information on this topic in the available literature?
Author Response
Thank you for your interesting and useful review of our manuscript.
Please see the attachment for our responses to your comments.
All the best,
Ingvil Laberg Holthe

Round 2
Reviewer 2 Report
Dear authors, thank you for addressing my concerns.
ps: p. 9, line 286: an error occurred during the revision process and 'in collaboration' in now written as 'inollaboration'.